# Trust, and distrust, of Ebola Treatment Centers: A case-study from Sierra Leone

**Paul Richards**[1]☯*, **Esther Mokuwa**[2]☯, **Pleun Welmers**[3]☯, **Harro Maat**[ID][3]‡, **Ulrike Beisel**[4]‡

**1** Directorate of Research and Planning, Njala University, Makonde, Sierra Leone, **2** Development Economics group, Wageningen University, Wageningen, The Netherlands, **3** Knowledge, Technology and Innovation group, Wageningen University, Wageningen, The Netherlands, **4** Department of Anthropology, University of Bayreuth, Bayreuth, Germany

☯ These authors contributed equally to this work.
‡ These authors also contributed equally to this work.
* paul.richards1945@gmail.com

**Data Availability Statement:** All relevant data are within the manuscript and its Supporting Information files.

**Funding:** This research was funded under the German Research Foundation Program "Infectious

## Abstract

The paper considers local responses to the introduction of an Ebola Treatment Centre in eastern Sierra Leone during the West African epidemic of 2014–15. Our study used qualitative methods consisting of focus groups and interviews, to gather responses from patients, members of the families of survivors and deceased victims of the disease, social liaison workers from the centre, and members of the general public. The data indicate that scepticism and resistance were widespread at the outset, but that misconceptions were replaced, in the minds of those directly affected by the disease, by more positive later assessments. Social workers, and social contacts of families with workers in the centre, helped reshape these perceptions, but a major factor was direct experience of the disease. This is apparent in the positive endorsements by survivors and families who had members taken to the facility. Even relatives of deceased victims agreed that the case-handling centre was valuable. However, we also present evidence of continuing scepticism in the minds of members of the general public, who continue to suspect that Ebola was a crisis manufactured for external benefit. Our conclusions stress the importance of better connectivity between communities and Ebola facilities to facilitate experiential learning. There is also a need to address the wider cognitive shock caused by a well-funded Ebola health initiative arriving in communities with a long history of inadequate health care. Restoring trust in medicine requires Ebola Virus Disease to be re-contextualized within a broader framework of concern for the health of all citizens.

## Introduction

The Ebola virus first became known to medical science in the 1970s after an outbreak of Ebola Virus Disease (EVD) in a mission hospital adjacent to the Ebola river in Zaire (now Democratic Republic of Congo [DRC]). Twenty or so episodes of EVD then followed, mainly in isolated areas of the central African forests. The exception was an urban episode in Gulu in northern Uganda in 2000. An EVD outbreak on the margins of the tropical forest belt in West

Diseases in Africa" with the grant number BE5682
4-1 (Grant holders: Beisel and Mokuwa). The
funders had no role in study design, data collection
and analysis, decision to publish, or preparation of
the manuscript.

**Competing interests:** The authors have declared
that no competing interests exist.

Africa in the Republic of Guinea, beginning in December 2013, rapidly turned into an epi-
demic, affecting three countries in particular–Guinea, Liberia and Sierra Leone [1].

A more recent outbreak in the provinces of North Kivu and Ituri in the Democratic Repub-
lic of Congo (DRC) in 2018 has potential to become as serious as the West African episode of
2013–15. The response in DRC is complicated by a long-running violent conflict in the region
and deep resentments against the state. Local populations are suspicious of Ebola responders
and violent acts have happened. For instance, in February 2019 two case-handling centres run
by the agency Medecins sans Frontieres (MSF) were attacked by armed militia [2]. Local com-
munities are suspicious and at times uncooperative [3–4]. To them, Ebola is only one of many
diseases they face. To some, the amount of attention paid to Ebola by government and interna-
tional responders suggests a hidden agenda; rumours are rife that the disease is not real, and a
political or money-making ruse [5]. This undermines the community cooperation needed for
infection control and suggests that more needs to be done to situate Ebola in a broader agenda
for health in marginalized, conflict-prone regions [4].

In this article we present work on community responses to EVD from the 2014–16 out-
break in Sierra Leone. Our research question was focused on how trust or distrust emerged in
outbreak control and Ebola treatment, and which factors are building or undermining trust in
medicine in Sierra Leone. Our data suggest that distrust between communities and Ebola
responders is a problem common to both epidemics. Given the North Kivu outbreak, the les-
sons of this West African study are of potential wider relevance. In particular, our material
highlights two key points. Firstly, we show that experiential learning and social feedback have
been crucial factors transforming initial distrust into trust–both with regards to knowledge
about Ebola as a disease itself, and to the acceptance of case-handling facilities (Ebola Treat-
ment Centres, henceforth ETC). We show this by tracing the opinions and experiences of sev-
eral stakeholders with one particular ETC in Sierra Leone, used here as a case study. Our paper
provides evidence that quarantine and isolation cut off social contact as an (unanticipated)
effect of medical concern and thereby distort important feedback mechanisms that would oth-
erwise allow communities to learn about Ebola.

This is a challenging lesson for developing future EVD interventions, because as we show in
this paper local communities are quick learners when directly confronted with Ebola (see also
[1]). Secondly, the paper argues that biomedical diagnosis and treatment differ significantly
from earlier patterns of health care, in particular in rural regions on the African continent.
Many Ebola patients did not have previous experience with blood-taking, and the workings of
biomedical diagnosis were unclear to them. If the diagnostic process and its epistemic presup-
positions are not carefully explained, this opens up the response to what is usually understood
as rumours and distrust. This is amplified by the fact that care in an ETC contrasts starkly with
previous health care patterns, where relatives play an important role in caring for patients. It is
thus crucial that patients and relatives leave the ETC with the experience they were well cared
for. Importantly, it is essential that health care systems cater for all diseases and not only for
epidemic emergencies of highly-infectious diseases, as a long-term priority for national and
international interventions [4, 6]. Trust in biomedicine is elusive if it is only striven for during
an epidemic. Indeed, in times of crisis, it is too late to lay the foundations for a relationship of
trust with biomedicine, or indeed to strengthen a health system that has been abandoned since
the advent of structural adjustment policies. In conclusion, preparedness must be integrated at
the level of ordinary health systems and well-functioning health systems are a necessary pre-
cursor to wide societal trust in biomedicine. Only if the national and international community
can show in practice that they are genuinely concerned with improving the long-term health
situation of populations will trust in biomedicine be sufficient to gain community support for
the stringent measures needed to tackle highly dangerous outbreaks of infection such as EVD.

## Materials and methods

The wider study on which this paper is based focused on the research question of how trust between health professionals and patients was built up or broke down in various organizational settings, in particular at the lower levels of the national health care delivery system in Sierra Leone, contrasting responses in urban and remote forest-edge rural locations. Some attention was also paid to Ebola-specific treatment facilities. Here we analyse a specific part of the data collected for the wider study, in particular a detailed interview -based case study of a case-handling facility close to Kenema, the regional capital of eastern Sierra Leone. The approach in this case study was qualitative. A mixed tool-set of focus group (Table 1) and in-depth key-informant interviews (n = 6) was used (see supporting materials for transcripts). All participants in interviews provided informed consent, and all responses have been rendered anonymous, including from a follow-up focus group held in a badly-affected village supplying patients to the case-handling facility, here given the fictional name of "Taninihun", and for which baseline questionnaire data and focus group materials were previously collected in December 2014, at the height of the epidemic (anonymised transcripts of these earlier focus—group and individual interviews are included in the supporting materials as a baseline for further focus-group reflections in 2017 by members of this village community).

Focus groups were carried out in July 2017 according to a method tested in earlier work on Ebola in Sierra Leone (protocol supplied as part of the supporting material). This makes use of a card tracking system to allow conversational voices to be identified and placed in order of the flow of the conversation without compromising the anonymity of the speaker. The system also has the advantage that it allows patterns to be detected in focus-group interaction–for instance, if there are dominant voices in the conversation, if certain comments only emerge after crucial interventions (e.g. prompts by facilitators), if particular themes emerge early or late in the conversation, if one age-group or gender is dominant, and so forth.

## Sampling and data analysis

The specific investigation of trust in the context of an Ebola case-handling facility (ETC) was intended to be an exemplary case-study, based on statements made in ten focus groups. The focus groups were run by a team of locally-based research assistants earlier trained by one of the authors (EYM) for work on Ebola in 2014–15. The same tailor-made focus-group interview protocol used in the 2014–15 study was again used in the research conducted in 2017 (see supporting materials).

**Table 1. Focus groups, membership, statements made.**

| Type of group | Number of persons | Statements made |
|---|---|---|
| Nurses | 7 | 29 |
| Dressers | 7 | 61 |
| ETC mixed group (sprayers, hygienists, supervisors) | 6 | 51 |
| Cooks | 5 | 37 |
| Social workers | 3 | 33 |
| Survivors | 2 | 36 |
| Families of survivors | 3 | 40 |
| Families of non-survivors | 4 | 60 |
| Kenema residents not directly affected | 6 | 51 |
| Residents of an affected village ("Taninihun") | 7 | 41 |
| **TOTAL** | **50** | **439** |

## Focus group participants

We identified participants in focus groups in the following manner. One of our research assistants had been employed in the case-handling facility in 2014, and helped us reach former employees, patients and their families. In all, fifty persons agreed to take part in our study, of which 28 were former employees of the case-handling facility, 10 were patients or members of families and affected communities, and six were members of the general public (Table 1). All gave informed consent by signing or marking a letter read out them describing the interview process and asking whether they wished to take part voluntarily.

The six persons representing the general public were selected randomly from urban neighbourhoods in Kenema. Randomization for membership of this group was done by raffle, first to choose six neighbourhoods and a street within each neighbourhood, then to choose one house on that street, and finally to determine the gender of the adult person to be interviewed in that house. All persons selected agreed to join the focus group. The districts from which they came were Kpetema, Fonikol, Buma 4, Kpayama II, Prison Barracks and Kenema Township.

Five focus groups were organised for occupational groups of ETC employees and three for survivors and family members. All persons contacted were invited, though not all agreed to take part.

A final (tenth) focus group was organised for randomly-selected villagers in "Taninihun", an Ebola-affected settlement where data were collected in December 2014 as part of a research study intended to support the epidemic-response social mobilization exercise [7]. Statements made in the 2017 village focus group were compared with statements made in three focus groups (for male elders, female elders, and younger people) held in December 2014, at the height of the epidemic. Additionally, we made reference to 25 randomly-sampled interviews with male household heads and adult women completed in "Taninhun" in December 2014, forming part of a gender-stratified national data set with sample size of over 700 [7].

All statements made by participants in the 2017 focus groups were written down by a facilitator designated for this task, and then entered into XL spread sheets, referenced according to the order in which remarks were made. The full set of statements is available in the on-line supporting materials. The authors classified statements by topic and used this as evidence for the points made in the paper.

## Ethics statement

Written consent for data collection was obtained from the Sierra Leone Ethics and Scientific Review Committee (SLESRC), which granted ethical approval to this study on 9 September, 2016. All adult subjects and signed or thumb-printed a letter read out to them telling them what the interview was about and what they should do if they felt uncomfortable. For the (few) interviewed teenage children the same procedure was followed involving the parent or guardian on the child's behalf.

## Background to the case study

The case-study ETC commenced activity in September 2014 and admitted its last case in March 2015. Since ETC were large facilities and there were only a handful in provincial Sierra Leone, there is no point in disguising the fact that this was the ETC located at Nganyahun, 10 miles north of Kenema, a case-handling facility established by the national Red Cross with support from the International Federation of Red Cross and Red Crescent Societies (IFRC). It was the first such centre to become operational in Sierra Leone. Nganyahun ETC drew on design principles developed by MSF [8]. In particular, use was made of a layout in which different

activities were assigned to "red" and "green" zones, of higher and lesser infection risk. Basically, activity always moved from the "green" to the "red" zone, and never the other way. Entrants to the "red" zone were required to wear full PPE, and those exiting this zone had to go through an extremely thorough and time-consuming decontamination process. Any departure from this protocol was met with sanctions. Repeat offenders were dismissed, for having put the safety of other staff at risk.

## Results

We focus here on the case study of the ETC in Nganyahun and retell the experiences of four key groups of people impacted by its presence: (i) a group constituted by people who lived through the Kenema outbreak but without being directly affected by it, (ii) ETC patients and relatives, (iii) ETC staff, including out-reach workers, and (iv) residents in "Taninihun", a village in Kenema District badly affected by Ebola, where patients and families had experience of both the ETC and the make-shift arrangements that preceded it. In what follows we report on the question of trust and distrust respectively, and how both conditions were emergent during the Ebola crisis. We have further sought to trace what role moments of experiential learning played in establishing and disseminating trust through social feedback processes.

### i. The view from the outside: Views on the ETC from the general public

In order to explore how the work in the ETC was seen from the general public surrounding the ETC, we interviewed a small group of residents of a mixed residential quarter in the nearby town Kenema, who were purposively selected on the basis of having lived in Kenema during the Ebola crisis but without having been directly affected by infection on a personal or family level. We formed a gender-mixed group of six people meeting these criteria and invited them to attend a focus group session (n = 6). The session facilitator offered various prompts, including a starting question about what members of the group thought were the causes of the Ebola outbreak. The explanation that the disease was spread through person-to-person contact was widely shared in the group. Only one person stated that wild animals may have been a factor, while implying this was not personal conviction ("we have been made to understand. . ."). At the time of the epidemic hunting and bush meat consumption had been a widely touted "official" explanation for the outbreak. However, this explanation was met with "epistemic dissonance" by locals who have long consumed bush meat without diseases occurring [9].

Nevertheless, most comments about the ETC itself were positive. Discussants had heard survivors speak, in person or over the radio, about the good care they received. This replaced pessimism based on earlier messages that there was no cure for Ebola with a growing confidence that many people could pull through with ETC help. The ETC also rapidly isolated EVD cases, and this reduced fear of visiting Kenema hospital, widely shunned as a result of a nosocomial outbreak in June-July 2014.

ETC staff were seen as bringing specialist knowledge, with benefits to patients, when compared to alternatives. Several discussants purported to have noticed that patients discharged from the ETC had fewer long-term health problems than those discharged from the government hospital. Two discussants even advocated for the ETC to be rebuilt, to cope with future infectious disease threats. Others, however, found the ETC premises a painful reminder of those they had lost, and were glad when it was dismantled. Thus, from the perspective of 2017, the ETC was generally seen in a positive light. But some members of the group did not hide earlier doubts. One offered a glimpse of those earlier assessments: "Most people were saying that [Ebola was] just a story for making money, and another frankly confessed: I was envious of their earnings."

There was also anger directed at some ETC employees, particularly the burial teams: "Even when they were highly paid, they still needed bribes before they could collect the dead bodies from the houses. Otherwise you were the last one they helped. I hated them so much." And lack of cultural sensitivity in burial also still rankled: "They were disrespecting the dead who were not buried properly, and I hated them for that reason."

A final and perhaps surprising note is the degree of awareness among our focus group that Ebola in Sierra Leone was not defeated by ETCs and foreign experts alone, but by a more general shift in attitudes in favor of reducing risks of bodily contact, by (for example) observing quarantine rules and local bye-laws. In response to a prompt about changing attitudes one discussant stated that: "Yes, [these] changed over time; the right treatment was now given to the patients and people were adhering by the by-laws." Another added: "We started obeying by-laws since those were ways to prevent the virus [from spreading] and as a result: infections were reduced." One young female discussant specifically noted that: "People changed their behavior and gave a personal example: my father even kept me in quarantine; I was not allowed to go to the market."

## ii. Survivors, families of survivors, and families of the deceased

A rather different picture emerges from the comments supplied by focus group participants who were survivors, or members of families of survivors and the deceased. Survivors and family members often had positive experiences of the ETC itself but suffered considerably from stigma and a lingering suspicion in their communities that the disease was fake.

Constituting a group of survivors to form a focus group posed a logistic challenge due to the way the ETC had received cases. Nganyahun ETC took a large proportion of its patients from outside the district, because by the time it was built the Kenema outbreak was ending. To follow up the group of survivors would have meant traveling to Freetown or Kono, the districts from which most patients came, and funds did not allow the team to travel, while so much of its other interview work was Kenema-based. Thus we had to locate a small group of Kenema-based survivors with experience of time spent in the ETC, and two of these agreed to discuss their experiences in depth. They both reflected on their fears before being taken to the ETC, having imbibed information that there was no cure, and hearing rumours that the ETC was involved in sinister activities. To their surprise they found that things were otherwise; they were well cared for and their personal needs were met–e.g. with a phone and credit to contact their families. One of the discussants summed it by saying: "All my perceptions were turned around, I was received well and well cared for immediately, what I heard is not what I saw." The other discussant added that care was meticulous, and it challenged his expectations: "Every hour people were checking on me; it proved to me that what people said about Ebola (that if you are taken to the ETC you will not survive) was all false."

There were some criticisms. Neither was told the result of their blood tests until discharge, and ambulance drivers and crew were castigated for rough driving and excessive spraying of chlorine.

One discussant also raised the issue that contradictory messages circulating about Ebola undermined confidence in treatment: "They [said] that there are no drugs for the disease, but when [I was] taken to the [ETC] I was given drugs. I feared to use those drugs [at that time]. That was a difficult situation." Both welcomed the fact that they had been given a discharge package to help with domestic reintegration, but they needed longer-term help, both for medical complications and with their life situation more generally (they had lost their jobs and struggled to pay children's educational expenses).

Both survivors report community alienation. One said: "We are seen by the society as false declarers of Ebola. We were not really expecting such, but [the] community still do not believe the existence of Ebola. Our relatives still [have] mixed feelings about us." The other added that: "The community [do] not believe a word we tell them. They have no trust in us about the existence of the virus." Additionally: "They thought that it was a bargain between us and the expatriates to make money. Even our former [friends who] we were with now isolate and stigmatize us."

The focus group on how families of survivors were affected comprised two women discussing husbands admitted to the ETC and a teenage girl referring to her father's time in the ETC. All three discussants thought the ETC was necessary and were grateful that staff had made such efforts to help their loved ones, despite the risk to their own lives.

All three spoke movingly about the pain of farewells and having to cope with fear that they would never see their family member again. By the time one of the patients was admitted (December 2014) the ETC had been operating for three months, and phones were now distributed to patients so that families could remain in touch. Even a certain amount of patient visiting seems to have been possible. One wife learnt her husband was going to survive when he was visited by his brother, who apparently talked to the sick man from behind a barrier.

One of the wives would have liked to help nurse her husband. The other recognized that because of the infection risk it was wiser to leave the task of nursing to skilled professionals. She, herself, was in quarantine, so had no opportunity to leave her house.

All three talked about being shunned by neighbors and friends. In one case neighbors even locked the local well so that it could not be used by the affected family. Experience of good care revised assessments of the ETC. Her view about the ETC changed: "Because there is now confidence that you will get well, if you do not hide your sickness and go for early treatment."

The four persons (two men, two women, three farmers and a trader, all middle aged or older) taking part in a focus group for those who had lost loved ones at the ETC provided nuanced assessments.

They all saw the establishment of the ETC as a reason for hope, and had expected their family member to survive, only to have these expectations dashed. Even so, they praised the dedication of the staff. One woman stated that: "I admire them as I speak because they did everything they could do to salvage the situation. Another added that: I am always happy when I see them, even though my husband died, my daughter survived. I have mixed feelings."

The pain of loss was made worse by uncomprehending friends and neighbors. One women said: "[I] was driven [away] by my husband when my son was infected and was taken to the ETC. I had to find another house to live in. I was marginalized by society." Another complained that: "People saw me as a bad person in the town because two persons [in] my family died of Ebola—my wife and my son-in-law. I was stigmatized." A third added: "We were very seriously mistreated, people never talked to us, and had nothing to do with me and my family. People didn't even come close to us. I was stigmatized."

Major complaints about the ETC centered on lack of provision for maintaining contact between patients and family. This began with the ambulance, which took no passengers: "When my wife was taken away, I felt discouraged. [I] wanted to have a word or two with her, but they denied me the privilege to go. The only thing that I did was to wave at her in the ambulance. My [wife] said to me, please take care of my children." In some cases, this was due to quarantine: "I was seriously disturbed when the ambulance took my husband and daughter to the ETC. I wept. I would have loved to [have] followed, but I was quarantined. They waved at me. I felt tears in my heart."

Over-zealous ambulance teams apparently added to the emotional distress by excessive use of chlorine spray: "When the ambulance arrived, they sprayed chlorine outside the house and

inside the ambulance and took my husband away. Every step he took was disinfected. I cried nearly to death."

A problem for families was that the ETCs had no capacity to allow family visits: "There was no visitation allowed. I cried my eyes out when I did not see my daughter." Another discussant added: "I felt so bad because I wanted to pay a visit to my husband but was not allowed; I was eager to go to see him but to no avail."

Next-of-kin did the next best thing and looked for relatives or contacts among the ETC staff. One said: "I had a kinsman from the same village (Taiama) who worked at the ETC, [and he] gave me information about what was happening at the ETC. I gave him my phone number [and] he used to call me and reported the death of my daughter." Another discussant "had the phone number of one of the workers [and] he used to give me whatever information I needed." But these back channels did not exist for everybody: "I had no relation with the ETC workers and therefore no information from anybody at the centre."

Family members felt especially upset that they could not contribute to the care of the sick person: "I was really dissatisfied when my husband was taken away from me and [I] was not given the privilege or chance to take care of him myself." A mother declared herself: "really disappointed and depressed that: there was no way I could take care of my daughter when she needed me most. Because I was not close [by] to launder or cook for her. I could not give her anything." A third informant commented that: "the workers in the ETC took care, but [they] will never do it well and in the same way as I would have done [it]."

Reporting of death is a sensitive and important cultural matter in Sierra Leone, surrounded by assumptions and protocols regarding both the messenger and when and to whom the message is to be conveyed. The ETC seemed not to have any systematic procedures for handling this. Much information was passed through informal channels by workers at the ETC. "A worker at the ETC (a cook) told me that my daughter was dead. When she saw my daughter at the ETC she recognized her, so when she died, she informed me." In another instance: "a worker from the ETC called one of the security personnel gating my quarantine to tell us my husband was dead." One husband reported that: "a member of the [Ebola] taskforce in the village told me that my wife was dead, and when I was asked further, he told me that this happened a week [previously]."

The topic of funerals aroused the greatest disquiet. A selection of comments include the following: "I felt bad because we never saw where they were buried. Who knows how they were buried? In a mass grave and denied our traditional requirements for a decent burial? [It is for that reason] that we bury our own deceased. We never set eyes on the corpse nor even visited the burial of our loved ones. We did not know there was such a ceremony. No, I did not see the corpse. I neither saw a picture of the corpse or of the grave. I did not know that there was a grave until [you told me] today."

There is in fact a well-organised Ebola cemetery at Nganyahun, with a large central memorial and marked and labelled burial plots, adjacent to the former ETC, but it is behind a high locked gate, and when one of us [PR] visited in 2016 we climbed the fence because no one could find the key.

One of the focus group respondents credited the ETC with taking some care over burial: "We were happy because at the ETC they take time to take care of the burial, the only [problem being] that we were not allowed to witness the burial." Another member knew about the graveyard since he criticized its positioning close to the wards: "the location of the cemetery close to the ETC was very unrealistic. [It served as] an agent of fear in the minds of the patients. It sent bad signals to both patients and family members."

The MSF-managed ETC at Bandajuma (Bo) took a more considerate approach, and organized burial in a special area for Ebola victims attached to the town's main public cemetery on the Dambala road.

### iii. ETC community liaison

The arrival of the international community to help with response to EVD, a disease never before seen in Sierra Leone, required abundant local help. Those with a good educational background found work as translators, case finders, data clerks and so forth. Some of these recruits were then trained as community liaison workers for the ETC. This group linked the facility with affected families and their communities, and also helped implement quarantine. Three of these liaison workers took part in a focus group to discuss their experiences. First, they talked about their initial fears and general ignorance of the disease.

Community liaison workers did not enter the ETC as part of their work, but at times came close to the fence, even at times apparently supplying food for patients to workers on the inside: "We rendered service to the patients, like getting close to the perimeter fence and giving them food." Presumably this was home cooking from families passed to workers inside the ETC.

In regard to the general ignorance of the disease, the learning curve of the liaison team was steep, as indicated in the following comments: "I was not trained at all before the ETC was established. But nobody knew much. By doing, we learnt [how to reduce] the death rate. One key thing I learnt is that when the outbreak occurred it was new and that we were not prepared, and that nobody [in the country] knew about the Ebola virus."

Part of the job was to help maintain connection between patients and their families cope and to explain the true nature of the disease: "[Through our work] relatives were directly connected to the patients. We spread messages that helped reduce the spread of Ebola." But misconceptions had to be overcome: "Low infected communities did not want to receive us. They did not believe in the disease. The only thing they believed is that if [Ebola] exist[ed], we had brought it to the community." One discussant thought that direct experience had changed attitudes: "[It] was totally different with the highly infected communities. They sympathized [with] us and [made] discharge[d] [patients] very welcome. Places like "Taninihun" [see below] received us well because of the deaths they [experienced] during the outbreak. They were anxious to get help."

A second assessment was more guarded: "Some communities welcomed us, but others were hostile. Sometimes people went on [the] rampage and threw stones at our vehicle. To my dismay up to now some people do not believe there was Ebola." Another added that: "in Kenema, people never appreciated us and had negative thoughts about us. They thought we were betraying them. They saw us as spies." Supervision of quarantine caused particular difficulty: "People felt we had been given money to keep them at home or deprive them [of] their freedom of movement. Some attacked us physically. They felt betrayed." Some of this suspicion was clearly connected to the vexed issue of burial practice: "Through the visits we made we came to realize that people had negative thoughts about us, [a] reason being that they thought we buried more than one dead body in a (mass) grave."

Misconception about the role of the ETC was also an acute problem initially: "[They thought] that [medical] people were giving infection to patients [so that they will] die and never come back again." Perception of the ambulance ride, with its over-use of chlorine, as a kind of one-way ticket to a death camp from which no one ever returned changed only with the discharge of survivors from about mid-October 2014: "[Once survivors] were discharged, and that was communicated through radio, there was more trust. In the end they had more trust [in the ETC] than in other health facilities."

### iv. "Taninihun" 2014 and 2017

The research team was directed by some of the ETC workers to "Taninihun" as a place where villagers had resisted sending patients to the ETC. This was denied by villagers, who pointed

out that their first cases dated back to July 2014, two months before the ETC was opened. A visit allowed us, however, to compare 2017 responses with the data collected in December 2014.

A focus group of six (three men and three women, five of whom described themselves as farmers) was formed, and participants unfolded a straightforward story. Yes, they had initially tried to care for Ebola victims at home: "There was no doubt about caring at home, because of symptoms of Ebola have existed before." Furthermore: "We did not believe the Ebola crisis was true until the cases worsened." Reluctance to report cases to the government hospital was due to a fear: "[that they would] remove their blood for sale [and that] having your blood removed for sale caused spread." Experience changed minds: "I only believed that Ebola existed after my husband was infected."

There were in all 28 cases in "Taninihun". When the community realized that Ebola was true, the chief together with all community members made by-laws, the main one being a fine for hiding a sick relative or [allowing] a stranger to pass the night: "these approaches we adopted for the protection of our village against Ebola." Quarantine and bye-laws ended the outbreak in the village, based on an understanding that spread was based on human-to-human contact: "Since the Ebola disease was not an airborne disease, but caused by body contact, [this] is one of the important factors that [made us] chose this approach."

With initial experience of Ebola care based on the government hospital [GH] the group was keen to compare this with what they knew about procedures at the ETC, a first indication they may have had closer contact with the latter than they were willing to admit: "The method of handling the dead at the ETC was better than the way the dead were handled at the GH, and no visitation was allowed at the GH, but visitation was allowed at the ETC and you could even send gifts for your loved ones at the ETC, [something] which was not permitted at the GH." Their regret now was that: "the ETC was not built early enough to save most of our people who died."

Reference back to the 2014 data set adds some important nuances, suggesting that the 2017 account has been modified to fit with a more general later understanding of the epidemic. According to the earlier data file there were 29 deaths and 13 survivors (so 42 cases of EVD in all). The first two persons to die were reportedly buried in the village, but the rest were interred in the graveyard attached to the Nganyahun ETC. The 2017 focus group implies that cases from "Taninihun" all went to the Kenema government hospital, but the 2014 source makes it clear this applied only to the two earliest cases, and that many (if not all) later cases from the village were handled by the ETC.

Thus, the outbreak in "Taninihun" almost certainly outlasted the cut-off date of August 2014 mentioned during the 2017 focus group session. The suggestion made by former employees of the ETC that villagers had experienced difficulty in accepting the facility evidently has some factual basis, as implicit in a statement made one villager in 2017, that liaison personnel from the ETC: "came [to "Tanininhun"] to persuade us more than six times."

## Discussion and conclusion

This paper has presented evidence that the first ETC in Sierra Leone, at Nganyahun, outside Kenema, was met with considerable initial suspicion and hostility. This distrust was connected to a more general climate of skepticism about the existence of Ebola as a disease. Much local reasoning supporting these negative attitudes is shared across widely separated epidemic locations in Africa. This places news from early 2019 that case-handling facilities in North Kivu have come under attack by armed militia in a significantly altered context. Doubtless, a complex insurgency complicates response efforts, but the underlying reasons for skepticism and

hostility lie in a negative dynamic linking Ebola responders and afflicted communities, or in other words in breaking "social accommodations" by crossing red lines [10].

Three major reasons can be identified for this lack of trust. The first reason is that Ebola (in both Sierra Leone and North Kivu) is a new disease, but it mimics the symptoms of many more familiar diseases, such as malaria and Lassa Fever. EVD reveals its distinctiveness only in its later stages. Isolated rural communities rightly see virtue in high-quality domestic care for other diseases such as malaria and feel deprived when they cannot offer the same for EVD. As the results of our case study shows, in Sierra Leone it took time for evidence to become clear to care givers. This temporal dynamic can be seen with regards to the fact that bodily contact spreads EVD, that quarantine and isolation are unavoidable, and that ETCs provide good quality care and can improve outcomes for patients.

Evidence was thus needed to support a change of attitudes in Sierra Leone. This emerged in three stages. The first was when families began to recognize that those most involved in care for a patient were those most at risk of next being infected. A steady flow of discharged survivors then changed perceptions that the ETC was a place where people went only to die. Families finally realized that patients had better survival chances in the ETC than those attending a regular hospital or kept at home. As other studies have also shown in times of emergency social learning has been fast, and many local practices were successfully adapted to meet newly acknowledged realities [1, 8, 11]. However, as we have shown here, practices of quarantine and isolation cut off social contact. The largely unanticipated effect of quarantine, however, was that important feedback mechanisms that would allow communities to learn about Ebola were disrupted, and so delayed learning based on the experience of trusted friends and family members. It furthermore created inequalities between those who knew ETC staff members and those who did not. It is in this situation that distrust and rumors found fertile ground to develop and spread. These findings underpin the importance of participatory mechanisms in learning [12, 13] that aims to build social structural support and collective efficacy when a community is confronted with health challenges [14].

The second reason for distrust is that the diagnosis of the disease is not readily understandable from prior local experience of health systems. The diagnosis of Ebola begins with a blood test. Phlebotomy is a well-known procedure to hospital patients in many parts of the world, but not in rural Africa, where reliable laboratories are few and far between. The most that many patients ever experience is a pin-prick blood test for malaria.

One puzzled village chief put the problem in these terms: "I have heard of giving a very sick patient a blood transfusion, but I have never heard of a very sick person being forced to give blood" [15]. He was describing the actions of an Ebola investigation team in collecting a blood sample from a dying woman. The team did not treat the woman, nor did they report the result of a blood test back to the chief. Only later was it apparent that the result must have been positive for Ebola when soldiers arrived in the village to quarantine it.

Such failures of communication about the basics of diagnosis appear to have been quite general in Sierra Leone. Survivors from the Kenema ETC told us that they were only told their diagnostic result (presumably Ebola negative) on being discharged. In this context, rumors that Ebola case handling centers were places for "mining" blood make some kind of sense, however wide of the mark. While these rumors are often dismissed as suspicions and "irrational" beliefs, it is important to recognize that many of such beliefs have a basis in colonial violence [16], as well as in ongoing unethical practices of blood and EVD sample extraction and theft in the context of the Ebola epidemic [17].

The third reason for lack of trust is closely related to the second reason in that the Ebola response hugely distorts the normal pattern of health care in poverty-stricken regions of Africa. This poses an explanatory challenge to local observers. Noting that everything was

focused on Ebola response one Sierra Leonean farmer asked us: what happened to all the other diseases, have they gone away? The question expresses disquiet that poverty-stricken Africa is left to cope with the other killer diseases largely unaided, while the world spares no expense to tackle Ebola.

This disparity raises doubts. A Sierra Leonean villager's saying is that if you see farmers running through a ripening field of rice you know they are running after, or running away, from something. So, what is it that the well-funded international Ebola response is running after, or running away from? Local answers vary, but all are disquieting. Perhaps foreigners are afraid this disease will affect them? Maybe there is money in body parts or blood? Or could the virus itself have some hidden utility (facts about Cold War germ warfare, and molecular patents, are not unknown in African villages).

It would be wrong to dismiss these stories as evidence of populist ignorance spread around by too easy access to cell phones and social media. It would be better to open up a conversation with patients, families and communities about "all the other diseases". Triaging and testing patients for Ebola offers an opportunity to treat other conditions that come to light [7]. As Nguyen [4: 1298] observes: "having patients emerge from isolation in improved health is powerful evidence that we aim to make everyone better, not just to stop Ebola's spread." In short, restoring trust in medicine requires Ebola Virus Disease to be re-contextualized within a broader framework of concern for the health of all citizens. In this sense we argue it is crucial that national and international community needs to evidence in practice that they are not only concerned with isolating a highly contagious disease outbreak, but are genuinely concerned with improving the long-term health situation of populations. It is only then that trust in biomedicine and its interventions can emerge sustainably.

## Supporting information

**S1 File. ETC focus groups.** Statements made by nine groups related to the ETC.
(XLSX)

**S2 File. Village focus group.** Statements made by "Taninihun" village group in 2017.
(XLSX)

**S3 File. Village questionnaire.** Results from the 2014 questionnaire.
(XLSX)

**S4 File. Interview transcripts.** Transcripts of six in-depth key-informant interviews.
(DOCX)

**S5 File. Focus group—Citizens group.** Statements made by randomly selected citizens.
(DOCX)

**S6 File. Tanininihun focus groups.** Statements made by "Taninihun" village group in 2014.
(DOCX)

**S7 File. Focus group protocol.**
(DOCX)

## Acknowledgments

First and foremost our heartfelt thanks go to our team of research assistants: Baigeh Johnson, Fomba Kanneh, Musa Konneh, Philip Lahai and Sao Bockarie. This study was funded by the German Research Foundation as part of the project "Trust in Medicine after the EVD epidemic: Street-level health bureaucrats, the institutionalization of care, and the creation of

preparedness in Sierra Leone, Uganda, and Ghana", we thank the DFG for the support. We also thank our collaborators in the project for intellectual companionship: Sung-Joon Park, Grace Akello, John Kuumuori Ganle, and Sylvanus Spencer. We thank Frederic Le Marcis for insightful comments as well as the two reviewers and editors of PLOS ONE for their constructive and helpful feedback.

## Author Contributions

**Conceptualization:** Paul Richards, Esther Mokuwa, Pleun Welmers, Harro Maat, Ulrike Beisel.

**Data curation:** Ulrike Beisel.

**Formal analysis:** Paul Richards, Esther Mokuwa, Ulrike Beisel.

**Funding acquisition:** Ulrike Beisel.

**Investigation:** Esther Mokuwa, Pleun Welmers.

**Methodology:** Paul Richards, Esther Mokuwa, Pleun Welmers, Harro Maat, Ulrike Beisel.

**Project administration:** Ulrike Beisel.

**Supervision:** Harro Maat.

**Writing – original draft:** Paul Richards, Pleun Welmers.

**Writing – review & editing:** Paul Richards, Esther Mokuwa, Harro Maat, Ulrike Beisel.

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
