## [Decision Letter · Decision Letter 0]

7 Aug 2019

PONE-D-19-15254

Trust, and distrust, of Ebola Treatment Centers: A case-study from Sierra Leone

PLOS ONE

Dear Dr Maat,

Thank you for submitting your manuscript to PLOS ONE. After careful consideration, we feel that it has merit but does not fully meet PLOS ONE’s publication criteria as it currently stands. Therefore, we invite you to submit a revised version of the manuscript that addresses the points raised during the review process.

We would appreciate receiving your revised manuscript by Sep 21 2019 11:59PM. To enhance the reproducibility of your results, we recommend that if applicable you deposit your laboratory protocols in protocols.io, where a protocol can be assigned its own identifier (DOI) such that it can be cited independently in the future. For instructions see: http://journals.plos.org/plosone/s/submission-guidelines#loc-laboratory-protocols

We look forward to receiving your revised manuscript.

Kind regards,

John Schieffelin, MD

Academic Editor

PLOS ONE

Journal Requirements:

1. Please provide the following information in your Methods section relating to your qualitative methodology: how many potential participants were approached, the expertise and training of the interviewers, and how the interviews or focus groups were recorded and transcribed.

2. Please include copies of the interview guide(s) used in the study, in both the original language and English, as Supporting Information, or include a citation if they have been published previously.

3. Thank you for including your ethics statement:

'Written consent for data collection was obtained from the Sierra Leone Ethics and Scientific Review Committee (SLESRC), 9 September, 2016. All adult subjects and signed or thumb-printed a letter read out to them telling them what the interview was about and what they should do if they felt uncomfortable. For the (few) interviewed teenage children the same procedure was followed involving the parent or guardian on the child’s behalf.'

Please amend your current ethics statement to confirm that your named institutional review board or ethics committee specifically approved this study.

Reviewers' comments:

Reviewer's Responses to Questions

**Comments to the Author**

1. Is the manuscript technically sound, and do the data support the conclusions?

Reviewer #1: Yes

Reviewer #2: Yes

2. Has the statistical analysis been performed appropriately and rigorously? 

Reviewer #1: N/A

Reviewer #2: N/A

3. Have the authors made all data underlying the findings in their manuscript fully available?

Reviewer #1: No

Reviewer #2: Yes

4. Is the manuscript presented in an intelligible fashion and written in standard English?

Reviewer #1: Yes

Reviewer #2: Yes

5. Review Comments to the Author

Reviewer #1: This paper presents results of a qualitative study of community perceptions of an Ebola treatment center in Sierra Leone during the 2014-2015 outbreak. The topic is timely given the challenges of containing the earlier W. Africa outbreak, as well as the significant barriers to responding to the current outbreak in DRC. The paper is well written and organized, and provides a compelling narrative of community perceptions and experiences. The authors also do a good job framing their study in the context of the current events in DRC. However, the paper also suffers from a variety of weaknesses which I enumerate below:

(1) The paper lacks discrete research questions. The narrative on pp. 3-4, where research questions could be included, read more of a summary of the study’s findings. To that end, lines 76-103 could be cut from this section as this material is more appropriate in a Discussion. Please add research questions, based on prior literature and current events, to guide the analysis and presentation of results.

(2) Methodological detail is scant and at times confusing:

-First, what is the larger study to which the authors refer? How is this current study distinct from the purpose(s) of the larger study?

-Second, the methods section makes reference to interviews, but the abstract only mentions focus groups. Please clarify.

-Third, the statement on line 113 (“The approach was largely qualitative”) raises questions – are quantitative data available? If so, will these data be used in the current study? Why or why not?

-Fourth, for focus groups conducted in both 2014 and 2017, how exactly were participants recruited?

-Fifth, the authors list a number of groups from which participants were purposively sampled, but how many participants were recruited within each group?

-Sixth, how were “ordinary citizens” randomly sampled? How did authors know they were “ordinary” and where did the authors locate these participants?

-Seventh, at the end of the “Sampling” section the authors revert back to discussion of the earlier 2014 study. Please find a way to describe each study succinctly and how the later study draws on the earlier one. The “Background to the case study” section could be repurposed to provide a linear presentation of the purpose and findings of the 2014 study, and how that study informs the current approach.

-Eighth, why were 25 interviews conducted in addition to the focus groups? As noted earlier for focus groups, how were these participants randomly sampled?

- Ninth, what did the interview and focus group protocols entail? Were these semi-structured, unstructured, etc? What are some examples of questions?

-Tenth, there is no section on analysis. How were these data analyzed? Were data coded? If so, how and by whom? What methodological approach did the authors follow?

(3) The results themselves are fascinating, albeit alarming. On line 224, the authors make this statement – “Getting a group of survivors to talk in a focus group was a challenge.” – but do not follow up. Why was this difficult and how did researchers approach this challenge? What are the implications of this challenge for the validity of the results?

(4) In general, in the Results section, I am confused about which results are derived from focus groups and which are derived from interviews. Please provide some additional structure to this discuss to clarify the sources of data being discussed.

(5) The Discussion is well done. This sentence particularly stands out to me: “In short, restoring trust in medicine requires Ebola Virus Disease to be re-contextualized within a broader framework of concern for the health of all citizens.” In my opinion, this issue is actually the backdrop for the entire study and should be emphasized more strongly as an organizing framework for the paper. The stigma around Ebola, the lack of understanding of transmission pathways, and the (understandable) mistrust of international actors all point to the failure of the larger health system to withstand the shock of Ebola – and the failure of the international health community to invest sufficiently in health systems strengthening versus (as the authors put it) situations where “foreigners are afraid this disease will affect them”.

Reviewer #2: This article is timely as the Ebola epidemic in DRC is a long-term one and technological responses (whether through vaccination or the use of individual isolation units) show the limits of their effectiveness. The authors' central argument is to emphasize (L80-82): "that experiential learning and social feedback have been crucial factors transforming initial distrust into trust - both with regards to knowledge about Ebola as a disease itself, and to the acceptance of Ebola Treatment Centres (ETC)”. However, while quarantine isolates the virus, it also makes it more difficult for information to flow between the confined and open worlds.

This argument is supported by the authors based on interviews conducted in 2017 regarding the installation of an ETU in a locality in Sierra Leone in 2014 and its functioning during the epidemic. These interviews were conducted with people of different ages, genders and status with the objective of cross-referencing the perceptions of ex-patients, families of patients (deceased or survivors) and social liaison workers about the ETUs.

The lesson is expensive but must be hammered again: organizing a response to an epidemic, whatever it may be:

- Cannot ignore taking into account the social and political context in which it is embedded;

- Cannot ignore that the response to the epidemic is interactive in nature: the terms of interaction between populations and actors in the response are not given but are built over time;

- Must be based on trust. This is a requirement for the establishment of a peaceful interaction, it is not decreed but is acquired largely through mutual learning that is built in experience and over time

The authors rightly underline the conditions of production of representations on the ETU, they insist on the necessary circulation of information between ETUs and the general population. They also report on the evolution of their appreciation throughout the epidemic as the quality of care improves and communication improves.

At the end of the reading of this text, some points deserve discussion. One concerns the international and national context of interaction, which is largely beyond the reach of field actors. The second concerns the role of actors who are the subject of little analysis and who are nevertheless crucial in the circulation of (relevant or non-relevant) information on ETUs, namely lay workers (hygienists, laundress women). Finally, the third concerns the complicated status of survivors and its impact on their ability to communicate efficiently.

An international and national context on which it is difficult to act

It is clear that the experience of the epidemic and its management depends on the quality of interactions between response actors and populations. This must be conducive to the establishment of a relationship of trust. However, the actors themselves have little means of intervening in the context of the interaction, whether it is international (African populations, however poorly educated they may be, make a pragmatic reading of the post-colonial relationships that structure their ordinary lives and in which the Ebola epidemic falls), or national: the illegitimacy of the actors in the response to the epidemic refers to the illegitimacy of national elites whose populations feel that they have long been forgotten. As is shown by the authors, people's distrust of the state and its proclaim goodwill does not begin with intervention.... Thus, the restoration of the confidence in biomedicine called for by the authors is illusory if it is only mentioned during Ebola. In times of crisis, it appears very late to lay the foundations for a relationship of trust with biomedicine (or to strengthen a health system that has been abandoned since the advent of structural adjustment policies...). Preparedness must also be considered at the level of ordinary health systems!

Finally, in the context of the DRC, it is necessary to reasonably question the very possibility of public health and even more so of research when the sources of legitimacy and power are not only multiple but also antagonistic (we think, for example, of the different militias in competition with the State).

Lay workers: underestimated informal communication actors:

The authors rightly point out that perceptions of ETU have evolved over time, over the course of the epidemic and through improved external communication of ETUs, better consideration of the expectations of patients and families, and evidence based on the experiences of patients and their families.

Concerning the communication of the ETU, an important group of actors remains ignored: they are the lay workers who have acted as hygienists, laundress. These actors are at the heart of the system, they work in large numbers in the ETUs and play a crucial role in the ETUs' informal communication. With low qualifications, they are in direct contact with ETU activities without necessarily understanding all their epidemiological and health dimensions. On the other hand, because of their social proximity to the general population, they are crucial channels of information between inside and outside. However, they are rarely explained by ETC caregivers. Formal communication within the ETUs should therefore be mobilized to serve as a basis for informal communication with the general population. This type of communication is undoubtedly the most effective because in a context of mistrust, it is one of the few that may not be perceived as remotely guided by the authority (political or health).

The ambivalent figure of the Ebola survivor

The relationship between populations and intervention is not necessarily fixed and the terms of interaction are constantly redefined. Credible testimony such as that of a survivor can thus have a positive effect on how the intervention is perceived. The impact of survivors can therefore be crucial, as the authors report. However, it remains to be recalled that the figure of the survivor is also the subject of negotiations. In Guinea during the 2013-2015 outbreak, they were sometimes perceived as having benefited from the intervention themselves, sometimes as having sold their own deceased parents and to benefit from Ebola resources (which, in the eyes of the populations, was confirmed by their position as official witnesses paid by NGOs). This figure of the survivor taking advantage of Ebola's money is not favorable to a positive reception of his speech. Survivors have often been ostracized by their relatives and the figure of a witness, the only recourse to exist for them after the sickness (as they were often excluded from their communities) masked the painful experience of the relegation that often characterizes survival after Ebola (Gomez-Temesio 2018).

Gomez-Temesio, Veronica 2018 “Outliving Death: Ebola, Zombies, and the Politics of Saving Lives: Outliving Death” American Anthropologist, 120 (4), 738-751, DOI: 10.1111/aman.1312

6. PLOS authors have the option to publish the peer review history of their article (what does this mean?). If published, this will include your full peer review and any attached files.

Reviewer #1: Yes: Thomas M. Crea

Reviewer #2: Yes: Frederic Le Marcis

---

## [Author Response · Author response to Decision Letter 0]

27 Sep 2019

Many thanks for soliciting reviewers for our piece and for their and your useful comments. We have revised the manuscript thoroughly and have done the following to address the requests:

General revisions requested by the editors and reviewers

1. Information on methodology: we have now included much more detail on our methodology: information on the wider study this manuscript emerged from; a clearly formulated research question; number of potential participants approached; training of the interviewers; a table detailing the status groups and numbers of research participants; information on sampling recording; transcription and analysis (as requested by the editors and reviewer 1)

2. Interview guides and data files: we have now included focus group guides, as well as the data from focus groups and from key-informants interview transcripts as supporting information. The questions asked in focus groups are listed in a column on each sheet of the focus group transcripts. 

3. Ethics statement: we have amended the ethics statement to confirm that the Sierra Leone Ethics and Scientific Review Committee has approved this study.

Specific issues raised by reviewer 1:

1. “Third, the statement on line 113 (“The approach was largely qualitative”) raises questions – are quantitative data available? If so, will these data be used in the current study? Why or why not?”

The bigger overarching study on trust in medicine after the Ebola epidemic in Sierra Leone, Ghana and Uganda largely utilized qualitative methods, but included a smaller quantitative survey in all three countries. The data analysed here, however, was collected in a different study period and with a slightly different thematic focus. In this manuscript we do not draw on the quantitative material, as it is not relevant to the topic of this manuscript.

2. “Getting a group of survivors to talk in a focus group was a challenge.” – but do not follow up. Why was this difficult and how did researchers approach this challenge? What are the implications of this challenge for the validity of the results?

We provide some additional information on this question in lines 367 – 373 of the manuscript. 

3. “In general, in the Results section, I am confused about which results are derived from focus groups and which are derived from interviews. Please provide some additional structure to this discuss to clarify the sources of data being discussed.”

The data in this paper are from focus group interviews and key informant interviews undertaken in 2017. All this material is now provided in the file of supporting material. Additionally we now explain that one of the focus group interviews in 2017 was conducted among villagers in a community (“Taninihun”) first studied at the height of the Ebola outbreak in 2014. We include in the supporting material questionnaire interview and focus group data from this earlier study as a “baseline” for the current analysis. Comparisons between the two data sets (for 2014 and 2017) are briefly discussed. But the paper primarily analyses the focus group and key-informant interview materials from 2017. Given the clarifications now added, we do not see a necessity to re-structure our results or discussion section. 

4. “This sentence particularly stands out to me: “In short, restoring trust in medicine requires Ebola Virus Disease to be re-contextualized within a broader framework of concern for the health of all citizens.” In my opinion, this issue is actually the backdrop for the entire study and should be emphasized more strongly”

Thank you for this point, we have emphasized it more throughout the manuscript.

Specific points raised by reviewer 2

1. “Thus, the restoration of the confidence in biomedicine called for by the authors is illusory if it is only mentioned during Ebola. In times of crisis, it appears very late to lay the foundations for a relationship of trust with biomedicine (or to strengthen a health system that has been abandoned since the advent of structural adjustment policies...). Preparedness must also be considered at the level of ordinary health systems!”

We thank the reviewer for this point and his clear phrasing of it. We have included it duly credited in the manuscript.

2. Lay workers: underestimated informal communication actors

We completely agree with this point. We have now included a table that details that our study included non-medical personnel at the ETUs, in particular dressers, cooks and social workers.

3. The ambivalent figure of the Ebola survivor

We completely agree with reviewer 2 on this point, but we believe that expanding our discussion on the ambivalence and stigma faced by survivors would distract from our core message. We agree that more research on survivors and their experiences is much needed, and we welcome the work done in the piece by Gomez-Temesio that reviewer 2 refers to. However, since this is not the main focus of our paper we have chosen to not expand on this point here. 

We would like to sincerely thank the reviewers and editors for their close and constructive engagement with our manuscript. We are convinced the revisions have helped the clarity and quality of the paper, and hope you find our revisions satisfactory. 

Kind regards,

---

## [Decision Letter · Decision Letter 1]

16 Oct 2019

Trust, and distrust, of Ebola Treatment Centers: A case-study from Sierra Leone

PONE-D-19-15254R1

Dear Dr. Maat,

We are pleased to inform you that your manuscript has been judged scientifically suitable for publication and will be formally accepted for publication once it complies with all outstanding technical requirements.

With kind regards,

John Schieffelin, MD

Academic Editor

PLOS ONE

Additional Editor Comments (optional):

Reviewers' comments:

Reviewer's Responses to Questions

**Comments to the Author**

1. If the authors have adequately addressed your comments raised in a previous round of review and you feel that this manuscript is now acceptable for publication, you may indicate that here to bypass the “Comments to the Author” section, enter your conflict of interest statement in the “Confidential to Editor” section, and submit your "Accept" recommendation.

Reviewer #1: All comments have been addressed

Reviewer #2: All comments have been addressed

2. Is the manuscript technically sound, and do the data support the conclusions?

Reviewer #1: Yes

Reviewer #2: Yes

3. Has the statistical analysis been performed appropriately and rigorously? 

Reviewer #1: Yes

Reviewer #2: N/A

4. Have the authors made all data underlying the findings in their manuscript fully available?

Reviewer #1: (No Response)

Reviewer #2: Yes

5. Is the manuscript presented in an intelligible fashion and written in standard English?

Reviewer #1: (No Response)

Reviewer #2: Yes

6. Review Comments to the Author

Reviewer #1: (No Response)

Reviewer #2: I consider the authors answered all the questions raised in both reviews and that the paper is now suitable for publication.

7. PLOS authors have the option to publish the peer review history of their article (what does this mean?). If published, this will include your full peer review and any attached files.

Reviewer #1: Yes: Thomas M Crea

Reviewer #2: Yes: Frederic Le Marcis, Pr of social anthropology, ENS de Lyon (Triangle UMR5206) & IRD (TransVIHMI

---

## [Editor Report · Acceptance letter]

15 Nov 2019

PONE-D-19-15254R1 

Trust, and distrust, of Ebola Treatment Centers: A case-study from Sierra Leone 

Dear Dr. Maat:

I am pleased to inform you that your manuscript has been deemed suitable for publication in PLOS ONE. Congratulations! Your manuscript is now with our production department. 

With kind regards,

on behalf of

Dr, John Schieffelin 

Academic Editor

PLOS ONE